# SEMI-SUPERVISED LONG-TAILED RECOGNITION USING ALTERNATE SAMPLING

## ABSTRACT

Main challenges in long-tailed recognition come from the imbalanced data distribution and sample scarcity in its tail classes. While techniques have been proposed to achieve a more balanced training loss and to improve tail classes data variations with synthesized samples, we resort to leverage readily available unlabeled data to boost recognition accuracy. The idea leads to a new recognition setting, namely semi-supervised long-tailed recognition. We argue this setting better resembles the real-world data collection and annotation process and hence can help close the gap to real-world scenarios. To address the semi-supervised long-tailed recognition problem, we present an alternate sampling framework combining the intuitions from successful methods in these two research areas. The classifier and feature embedding are learned separately and updated iteratively. The class-balanced sampling strategy has been implemented to train the classifier in a way not affected by the pseudo labels' quality on the unlabeled data. A consistency loss has been introduced to limit the impact from unlabeled data while leveraging them to update the feature embedding. We demonstrate significant accuracy improvements over other competitive methods on two datasets.

## 1 INTRODUCTION

Large-scale datasets, which contain sufficient data in each class, has been a major factor to the success of modern deep learning models for computer vision tasks, such as object recognition. These datasets are usually carefully curated and balanced to have an uniform data distribution over all classes. This balanced data distribution favors model training but could be impractical in many real world applications, where the frequency of samples from different classes can be imbalanced, leading to a long-tailed data distribution. As shown in Figure 1(b), several highly populated classes take up most of the labeled samples, and some of the classes only have very few samples during training.

The long-tailed recognition problem has been widely studied in the literature. One major challenge in this setting (Liu et al. (2019); Kang et al. (2020); Zhou et al. (2020)) to deep learning model training is the tendency of under-fitting in less-populated classes. The root causes of this under-fitting are the imbalanced training data distribution as well as the scarcity of data samples in the tail classes.

More specifically, with an imbalanced training data distribution, when several head classes take up most of the training samples, tail classes contribute little in the training loss. The model is such that biased towards head classes. Prior works (Lin et al. (2017); Cao et al. (2019); Kang et al. (2020); Zhou et al. (2020)) tried to mitigate the issue by re-sampling the training data to be a balanced distribution or calibrating the sample weights in calculating the loss. However, still the scarcity of tail class data samples limits the intra-class variations and overall recognition accuracy. Methods focusing on few-shot learning have been introduced to address this problem through data augmentation and data synthesis (Wang et al. (2018); Hariharan & Girshick (2017); Liu et al. (2020)).

In this work, we resort to a different path to leverage massive unlabeled real data in training to help improve the long-tailed recognition accuracy. Since data collection is much cheaper and accessible comparing to data annotation, additional unlabeled real data could readily be available in many real-world scenarios. This semi-supervised learning setting has been intensively studied in the literature (Laine & Aila (2016); Rasmus et al. (2015); Tarvainen & Valpola (2017); Berthelot et al. (2019);

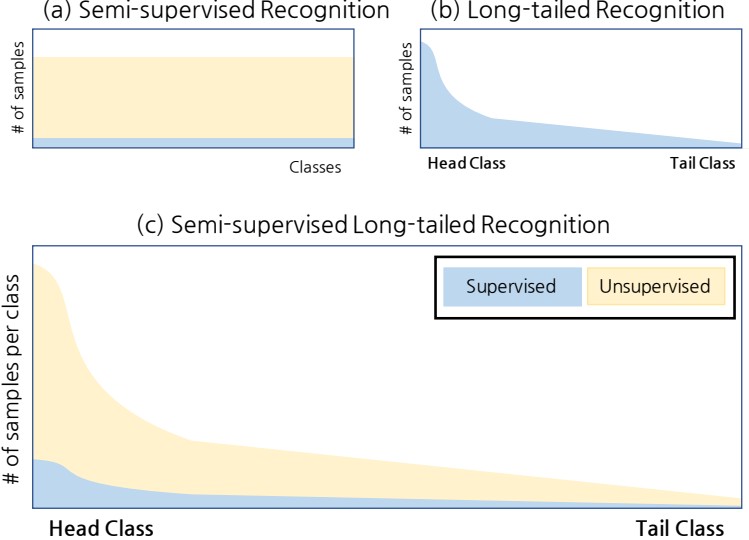

Figure 1: Comparison of different recognition paradigms: a) statistics of CIFAR-10 when used as a Semi-supervised Recognition benchmark; b) typical data distribution over classes in long-tailed recognition; c) the proposed Semi-supervised long-tail recognition setting, in which both labeled and unlabeled subsets follow the same underlying long-tailed data distribution.

Sohn et al. (2020)). However, as shown in Figure 1(a), when we carefully look at the data distribution of the widely used benchmarks, we observe well-balanced labeled subset and unlabeled subset. As discussed above, the manually curated balanced distribution, can lead to a gap to real-world scenarios. This is especially true in unlabeled data. Without labels, people have no way to balance the data among classes.

In this paper, we propose a more realistic and challenging setting, namely semi-supervised long-tailed recognition. As shown in Figure 1(c), we assume a long-tailed data distribution of the overall dataset and both the labeled and unlabeled subsets of training data follow the same underlying long-tailed data distribution. This setting generally resembles a realistic data collection and annotation workflow. After collecting the raw data, one has no knowledge of its class distribution before annotation. As it is expensive to annotate the full corpus, a common practice is to randomly sample a subset for annotation under a given labeling budget. When the raw data follows a long-tailed class distribution, we should expect the same in the labeled subset.

While this new recognition paradigm shares the challenges in both semi-supervised learning and long-tailed recognition, there is no readily naive solution to it. Methods in long-tailed recognition rely on class labels to achieve balanced training, which are not available in the unlabeled portion in the semi-supervised long-tailed recognition. Prior semi-supervised methods without considering the long-tailed distribution could fail as well.

Taking one of the competitive baseline methods for example, (Yang & Xu (2020)) proposed to firstly train a recognition model with the labeled subset to generate pseudo labels for the unlabeled subset, then the model is fine-tuned with the full training dataset. However, when the labeled subset follows a long-tailed distribution, the pseudo labels are much less accurate for tail classes than head classes. As a result, the overall pseudo labels quality could be too bad to leverage (See Section 4.4 for results in CIFAR-10-SSLT and ImageNet-SSLT).

To address the semi-supervised long-tailed recognition problem, we present a method designed specifically for this setting. We bring the successful class-balanced sampling strategy and combined it with model decoupling in an alternate learning framework to overcome the difficulty of balancing unlabeled training data.

Inspired by (Kang et al. (2020)), we decouple the recognition model into a feature embedding and a classifier, and train them with random sampling and class-balanced sampling respectively. As we

are targeting at a semi-supervised setting, the classifier is only trained on labeled data to get around the difficulty of applying correctly class-balanced sampling on unlabeled data, aligning with the intuition that the classifier needs more robust supervision than the feature embedding.

After that, with the proposed alternative learning framework, we improve model by updating the feature embedding and the classifier iteratively. We assign pseudo labels with the up-to-date classifier and observed gradually improved accuracy of pseudo labels over iterations. The pseudo labels are then incorporated in fine-tuning the feature embedding with a regularization term to limit its potential negative impacts. Similar iterative design has been proposed in semi-supervised learning literature (Laine & Aila (2016); Tarvainen & Valpola (2017)) but important implementation details differ.

To summarize, in this paper, 1) we resort to semi-supervised learning to help improve long-tailed recognition accuracy and identify practical gap of current semi-supervised recognition datasets due to their well-balanced unlabeled subset; 2) we propose a new recognition paradigm named semi-supervised long-tailed recognition better resembling real-world data collection and annotation workflow; 3) we propose a new alternative sampling method to address the semi-supervised long-tailed recognition and demonstrate significant improvements on several benchmarks.

## 2 RELATED WORK

**Long-tailed recognition** has been recently studied a lot (Wang et al. (2017); Oh Song et al. (2016); Lin et al. (2017); Zhang et al. (2017); Liu et al. (2019); Wang & Hebert (2016)). Several approaches have been proposed, including metric learning (Oh Song et al. (2016); Zhang et al. (2017)), loss weighting (Lin et al. (2017)), and meta-learning (Wang & Hebert (2016)). Some methods design dedicated loss functions to mitigate the data imbalanced problem. For example, lift loss (Oh Song et al. (2016)) introduces margins between many training samples. Range loss (Zhang et al. (2017)) encourages data from the same class to be close and different classes to be far away in the embedding space. The focal loss (Lin et al. (2017)) dynamically balances weights of positive, hard negative, and easy negative samples. As reported by (Liu et al. (2019)), when applied to long-tailed recognition, many of these methods improved accuracy of the few-shot group, but at the cost of lower accuracy over the many-shot classes.

Other methods, e.g. LDAM-DRW (Cao et al. (2019)) replace cross-entropy loss with LDAM loss. This adds a calibration factor to the original cross-entropy loss. When combined with loss re-weighting, it improves the accuracy in all splits in long-tailed recognition. However, it can not be easily generalized to semi-supervised learning. Because both the calibration factor and the loss weight are calculated based on the number of samples of each class.

In face recognition and person re-identification, the datasets are mostly with long-tailed distribution. LEAP (Liu et al. (2020)) augmented data samples from tail (few-shot) classes by transferring intra-class variations from head (many-shot) classes. Instead of data augmentation, we introduce unsupervised data to improve the performance of long-tailed recognition.

A recent work (Yang & Xu (2020)) rethinks the value of labels in imbalance learning. As part of the discussion, semi-supervised learning is included. However, only the basic pseudo label solution and simple datasets, such as CIFAR and SVHN, are discussed.

More recent works (Kang et al. (2020); Zhou et al. (2020)) with improved long-tailed recognition share the observation that feature embedding and the classifier should be trained with different sampling strategies. In this work, we adopt our method on this observation to learn the feature embedding model with random sampling and train the classifier with class-balanced sampling. This design is further closely compatible with semi-supervised learning under alternate learning.

**Semi-supervised learning** has been extensively discussed in recognition discipline (Laine & Aila (2016); Rasmus et al. (2015); Tarvainen & Valpola (2017)). One common observation is to optimize the traditional cross-entropy loss together with a regularization term that regulates the perturbation consistency of unlabelled data.

Ladder net (Rasmus et al. (2015)) is introduced to minimise the reconstruction loss between the network outputs from a given sample and its perturbation. It is then simplified in (Laine & Aila (2016)) as two temporal modules: Π-Model and Temporal Ensembling. The Temporal Ensembling

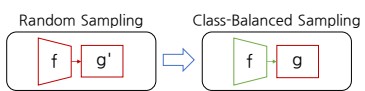

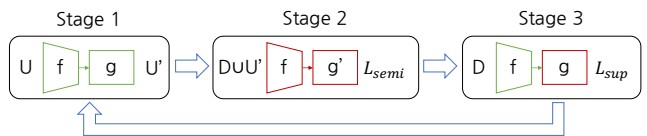

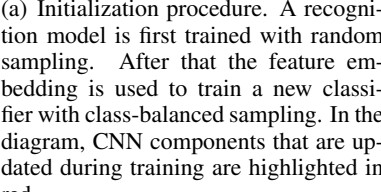

(a) Initialization procedure. A recognition model is first trained with random sampling. After that the feature embedding is used to train a new classifier with class-balanced sampling. In the diagram, CNN components that are updated during training are highlighted in red.

(b) Diagram of alternate learning. CNN modules in green line is only used in forwarding. Those in red are fine-tuned with the corresponding loss. In Stage 1, samples from $\mathcal{U}$ are forwarded through $f$ and $g$. $\mathcal{U}'$ consists of samples from $\mathcal{U}$, and pseudo labels acquired from $g$. In Stage 2, $f$ and $g'$ are trained on the combination of $\mathcal{D}$ and $\mathcal{U}'$. In Stage 3, only the classifier $g$ is trained. $f$ is fixed and only used in forwarding.

encourages the output of the network from unlabeled data to be similar to its counterpart from previous training epoch. More recently, Mean Teacher (Tarvainen & Valpola (2017)) extends it by assembling along training. Instead of storing previous predictions, they assemble a Teacher model by calculating the moving average of the training network, i.e. the Student. The Teacher is then used to provide the consistency of predictions to the Student.

In addition to that, MA-DNN (Chen et al. (2018)) introduces a memory module to maintain the category prototypes and provide regularization for learning with unlabeled data. Label propagation (Li et al. (2018)) is also considered with the help of label graph. More recently, Mixmatch (Berthelot et al. (2019)) and Fixmatch (Sohn et al. (2020)) improve the performance by introducing powerful data augmentations and perturbation consistencies.

All the semi-supervised methods above do not separate labeled data during semi-supervised training. In fact, it is beneficial to combine labeled data and unlabeled data in a certain proportion (Laine & Aila (2016); Tarvainen & Valpola (2017)). However, without further knowledge, we have no insight how to deal with this combination when long-tailed distribution is included. Furthermore, long-tailed learning methods require calibration or re-sampling based on the class distribution. This combination of labeled and unlabeled data makes the distribution unstable. In result, this is not suitable for long-tailed recognition.

Recently, Salsa (Rebuffi et al. (2020)) proposes to decouple the supervised learning from semi-supervised training. Our method follows the alternate training scheme from it, because it is surprisingly compatible with long-tailed learning. In practice, our method differs from Salsa in the following aspects.

First, we adopt class-balanced sampling in supervised learning to deal with the long-tailed distribution. Second, we use supervised learning instead of self-supervised learning as initialization. We find that self-supervised learning results in inferior performance in long-tailed scenario. Third, the re-initialization is not needed. Because our initialization is already from supervised learning, there is not a specific starting point to re-initialize the model. In fact, this enhances the soft constraint between the two stages in (Rebuffi et al. (2020)).

With the models continuously optimized along alternate learning, our method achieves superior performance while maintains the same amount of training epochs as fine-tuning on pseudo labels.

## 3 METHOD

In this section, we will introduce the proposed method to semi-supervised long-tailed recognition. The semi-supervised long-tailed recognition problem is first defined, and some notations are clarified. The decoupling strategy of long-tailed recognition is then discussed. This is also the initialization phase of our method. After that, the alternate learning scheme with 3 stages is fully discussed.

### 3.1 SEMI-SUPERVISED LONG-TAILED RECOGNITION

We start by defining the semi-supervised long-tailed recognition problem. Consider an image recognition problem with a labeled training set $\mathcal{D} = \{(\mathbf{x}_i, \mathbf{y}_i); i = 1, \ldots, N\}$, where $x_i$ is an example

and $y_i \in \{1, \ldots, C\}$ its label, where $C$ is the number of classes. For semi-supervised learning, there is also an unsupervised training subset $\mathcal{U} = \{\mathbf{x}_i; i = 1, \ldots, M\}$.

Although the labels of data in $\mathcal{U}$ are not available, every sample has its label from $\{1, \ldots, C\}$. For class $j$, we have $n_j$ samples from $\mathcal{D}$ and $m_j$ samples from $\mathcal{U}$. With the assumption that supervised and unsupervised data follow the same distribution, we have the fact $\frac{n_j}{N} = \frac{m_j}{M}, \quad \forall j$.

The testing set, on the other hand, in order to evaluate the performance on every class without bias, is balanced sampled on all classes in $\{1, \ldots, C\}$.

## 3.2 MODEL DECOUPLING AND DATA SAMPLING

A CNN model combines a feature embedding $\mathbf{z} = f(\mathbf{x}; \theta) \in \mathbb{R}^d$, and a classifier $g(\mathbf{z}) \in [0, 1]^C$. Embedding $f(\mathbf{x}; \theta)$ is implemented by several convolutional layers of parameters $\theta$. The classifier operates on the embedding to produce a class prediction $\hat{y} = \arg\max_i g_i(\mathbf{z})$. In this work, we adopt the popular linear classifier $g(\mathbf{z}) = \nu(\mathbf{W}\mathbf{x} + \mathbf{b})$, where $\nu$ is the softmax function.

Standard (random sampling) training of the CNN lies on mini-batch SGD, where each batch is randomly sampled from training data. A class $j$ of $n_j$ training examples has probability $\frac{n_j}{N}$ of being represented in the batch. Without loss of generality, we assume classes sorted by decreasing cardinality, i.e. $n_i \leq n_j, \forall i > j$. In the long-tailed setting, where $n_1 \gg n_C$, the model is not fully trained on classes of large index $j$ (tail classes) and under-fits. This can be avoided with recourse to non-uniform sampling strategies, the most popular of which is class-balanced sampling. This samples each class with probability $\frac{1}{C}$, over-sampling tail classes.

Kang et al. (2020); Zhou et al. (2020) shows that while classifier benefits from class-balanced sampling, feature embedding is more robust in random sampling. Practically, Kang et al. (2020) achieves this by decoupling the training into two stages, and train the feature embedding with random sampling in the first stage, and classifier the second with class-balanced sampling.

## 3.3 INITIALIZATION

The initialization of the proposed method follows the decoupling from (Kang et al. (2020)). The two-stage initialization is illustrated in Figure 2(a). A CNN model is first trained with random sampling. A feature embedding $\mathbf{z} = f(\mathbf{x}; \theta) \in \mathbb{R}^d$, and a classifier $g'(\mathbf{z}) \in [0, 1]^C$ are acquired. After convergence, the classifier is re-initialized and trained with class-balanced sampling, with the feature embedding fixed. This results in a class-balanced classifier $g(\mathbf{z}) \in [0, 1]^C$. Both the feature embedding and the classifier are trained on the supervised training subset $\mathcal{D}$.

## 3.4 ALTERNATE LEARNING

After obtaining an initialized model, most semi-supervised learning methods fine-tune the model on a combination of supervised and unsupervised samples. This is, however, incompatible with our long-tailed recognition model. When applied on unsupervised data, we have no ground truth for class-balanced sampling. One can make a sacrifice by relying on pseudo labels assigned by the initialized model. But the effectiveness will depend on the accuracy of pseudo labels.

It is even worse when considering the fact that long-tailed models usually have better performance on highly populated classes and worse on few-shot classes. Class-balanced sampling over-samples few-shot classes, while down-samples many-shot. This means, in general, the worse part of pseudo labels contributes more to the training loss than it should be, while the better part contributes less.

Another difficulty is the model compatibility when combining the long-tailed model to semi-supervised learning methods. Many semi-supervised learning methods evolve the model and pseudo labels at the same time. For example, Mean Teacher (Tarvainen & Valpola (2017)) assembles the teacher model by moving average and trains the student with consistency loss. When it comes to long-tailed model, it is not clear when we should update the feature embedding or classifier. And it is also difficult to incorporate both random and class-balanced sampling.

Inspired by (Rebuffi et al. (2020)), which separates supervised learning apart from semi-supervised learning, we propose an alternate learning scheme. The supervised training on data $\mathcal{D}$, and semi-

supervised training on data $\mathcal{D} \cup \mathcal{U}$ are carried out in an alternate fashion together with model decoupling and different data sampling strategies.

In practice, after initialization, we have a feature embedding $\mathbf{z} = f(\mathbf{x}; \theta)$, a classifier $g'(\mathbf{z})$ trained with random sampling, and a classifier $g(\mathbf{z})$ trained with class-balanced sampling. In (Kang et al. (2020)), only $g(\mathbf{z})$ is used in testing. However, we keep the randomly trained classifier $g'(\mathbf{z})$ for further usage. The training scheme iterates among 3 stages for $N$ loops, which are shown in Figure 2(b).

**Stage 1: Label assignment.** In this stage, pseudo labels are assigned for the unsupervised subset $\mathcal{U}$. The feature embedding $f(\mathbf{x}; \theta)$ and class-balanced classifier $g(\mathbf{z})$ are used. The choice of classifier is equivalent to the long-tailed model when tested for better overall accuracy. The unsupervised subset with pseudo labels is $\hat{\mathcal{U}} = \{(\mathbf{x}_i, \hat{\mathbf{y}}_i); i = 1, \ldots, M\}$, where $\hat{\mathbf{y}}_i$ are pseudo labels.

**Stage 2: Semi-supervised training.** After label assignment, we have pseudo labels for all unsupervised data. The model is the fine-tuned on the combination of true and pseudo labels, i.e. on $\mathcal{D} \cup \hat{\mathcal{U}}$. In this stage, random sampling is used to update the feature embedding $f(\mathbf{x}; \theta)$ and the randomly-trained classifier $g'(\mathbf{z})$. The classification is optimized by cross-entropy loss:

$$\mathcal{L}_{CE} = \sum_{(\mathbf{x}_i, y_i) \in \mathcal{D} \cup \hat{\mathcal{U}}} -\log g'_{y_i}(f(\mathbf{x}_i; \theta)), \tag{1}$$

where $g'_{y_i}$ is the $y_i$-th element of $g'$.

In semi-supervised learning literature, a regularization loss is usually applied to maintain the consistency for unlabeled data. This consistency loss captures the fact that data points in the neighborhood usually share the same label. We adopt this idea and implement the temporal consistency from (Laine & Aila (2016)). In practice, the class probabilities are acquired from $g'$. Given the class probability $p^{e-1}$ from epoch $e-1$, and the class probability $p^e$ from epoch $e$, the loss is KL-divergence between the two.

$$\mathcal{L}_{consist} = \sum_{(\mathbf{x}_i, y_i) \in \mathcal{D} \cup \hat{\mathcal{U}}} \sum_j p_j^{e-1} \log \frac{p_j^{e-1}}{p_j^e}, \tag{2}$$

where $p_j^{e-1}$ and $p_j^e$ are the $j$-th element of $p^{e-1}$ and $p^e$ respectively.

Overall, the semi-supervised learning loss is the combination of the two.

$$\mathcal{L}_{semi} = \mathcal{L}_{CE} + \lambda \mathcal{L}_{consist}. \tag{3}$$

**Stage 3: Supervised training.** We update the class-balanced classifier $g(\mathbf{z})$ based on the refined feature embedding, which is fine-tuned with semi-supervised learning in Stage 2. Specifically, the fine-tuning is applied with class-balanced sampling and only on the supervised subset $\mathcal{D}$. In this stage, only classifier is updated. The feature embedding is fixed and only used in forwarding. Given the class-balanced version of supervised subset $\mathcal{D}'$, the cross-entropy loss for classification is

$$\mathcal{L}_{sup} = \sum_{(\mathbf{x}_i, y_i) \in \mathcal{D}'} -\log g_{y_i}(f(\mathbf{x}_i; \theta)), \tag{4}$$

where $g_{y_i}$ is the $y_i$-th element of $g$.

### 3.5 INSIGHT OF THE DESIGN

**Feature embedding** is trained with random sampling and semi-supervised learning. This is consistent with long-tailed model in the sampling scheme. It also follows the fact that feature embedding is less prone to noisy labels. Actually, in self-supervised learning literature (Gidaris et al. (2018); He et al. (2020); Chen et al. (2020)), the feature embedding can even be learned without labels.

**Classifier** is learned with class-balance sampling and only supervised data. This is again the same as the supervised version. And by avoiding fitting the classifier on pseudo labels, we prevent the wrong labels from propagating through the whole training process. Given the fact that the pseudo labels are provided by the classifier, if classifier is still optimize on those, wrong labels can be easily maintained in the fine-tuned version of the classifier.

Table 1: Left: Results(Accuracy in %) on CIFAR-10-SSLT. ResNet-18 is used for all methods; Right: Results(Accuracy in %) on ImageNet-SSLT. ResNet-18/50 are used for all methods.

| | Imbalance factor=100/1000 | | | | ResNet-18/50 | | | |
| Method | Overall | Many-Shot | Medium-Shot | Few-Shot | Overall | Many-Shot | Medium-Shot | Few-Shot |
|---|---|---|---|---|---|---|---|---|
| LDAM-DRW (L) | 67.4/46.2 | 79.7/70.3 | 54.2/36.3 | 68.1/35.6 | 21.3/24.9 | 42.6/51.2 | 27.0/31.1 | 8.6/9.9 |
| Pseudo-Label + L | 69.6/48.4 | 69.7/74.0 | 55.1/39.3 | 80.2/36.0 | 17.6/23.9 | 22.4/44.0 | 20.9/30.0 | 12.6/11.1 |
| Mean Teacher + L | 69.9/48.3 | 69.7/75.7 | 57.3/41.4 | 79.4/32.9 | 21.3/25.6 | 41.8/49.1 | 28.1/31.8 | 7.6/11.7 |
| Decoupling (D) | 64.0/45.8 | 91.1/86.5 | 63.0/47.2 | 44.4/14.4 | 24.8/27.2 | 53.9/58.5 | 31.1/34.2 | 8.7/9.8 |
| Pseudo-Label + D | 68.9/46.5 | 92.7/89.0 | 70.8/47.0 | 49.8/14.2 | 25.3/27.7 | 47.6/52.2 | 32.1/34.7 | 11.1/12.4 |
| Ours | 71.3/66.7 | 89.5/84.4 | 67.7/69.4 | 60.2/51.4 | 26.5/29.0 | 52.0/57.1 | 33.9/36.5 | 10.7/12.3 |

Training the classifier only on labeled data also avoids the dilemma of class-balancing on unlabeled data. Without ground truth labels, class-balanced sampling can only rely on pseudo labels, which are not perfect. And the fact that pseudo labels have more errors on few-shot classes is specially not suitable for class-balanced sampling. Because when few-shot classes are over-sampled, those errors are also scaled up during training.

## 4 EXPERIMENTS

### 4.1 DATASETS

We manually curate two semi-supervised long-tailed recognition benchmarks.

**CIFAR-10-SSLT.** For easy comparison and ablation, we compose a lightweight semi-supervised long-tailed dataset based on CIFAR-10 (Krizhevsky et al. (2009)). Following (Cao et al. (2019)), we randomly sample the training set of CIFAR-10 under an exponential function with imbalance ratios in $\{100, 1000\}$ (the ratio of most populated class to least populated). The unsupervised subset is collected from Tiny Images (Torralba et al. (2008)) following the strategy introduced in (Yang & Xu (2020)). The class distribution of unlabeled data is always the same as the labeled one, with 5 times larger. For better description and comparison, we assign the 10 classes into 3 splits: many-shot, medium-shot, few-shot, with many-shot the most populated 3 classes, medium-shot the medium 3, and few-shot the least 4 classes.

**ImageNet-SSLT.** To evaluate the effectiveness of semi-supervised long-tailed recognition methods on large-scale datasets, we assemble a challenging dataset from ImageNet (ILSVRC-2012) (Deng et al. (2009)). The supervised subset is sampled with Lomax distribution with shape parameter $\alpha = 6$, scale parameter $\lambda = 1000$. It contains $41,134$ images from 1000 classes, with the maximum of 250 images per class and the minimum of 2 samples. The unsupervised subset is sampled under the same distribution with an unsupervised factor 4, i.e. $|\mathcal{U}| = 4|\mathcal{D}|$. The 1000 classes are divided into 3 splits based on the amount of labeled data $n$: many-shot ($n > 100$), medium-shot ($10 < n \leq 100$), few-shot ($n \leq 10$). In result, the dataset has 140 many-shot, 433 medium-shot, and 427 few-shot classes. Methods are evaluated under all classes and each class split.

### 4.2 NETWORK ARCHITECTURE

ResNet-18 (He et al. (2016)) is used on both CIFAR-10-SSLT and ImageNet-SSLT for fast experiments and comparison. ResNet-50 is used on ImageNet-SSLT to show how methods scale up to larger networks.

### 4.3 COMPARISON METHODS

To our best knowledge, there is no available method designated for semi-supervised long-tailed recognition. We explore typical long-tailed recognition methods and semi-supervised recognition methods, and combine them as baselines.

**Long-tailed Recognition.** We consider two long-tailed methods, one for loss calibration and the other for re-sampling. LDAM-DRW (Cao et al. (2019)) converts cross-entropy loss to LDAM loss with calibration factors based on class counts. It further regulates the loss with a loss weight also from class counts. Decoupling (Kang et al. (2020)) decouples the training of embedding and classifier with different sampling strategies. This is also the initialization in our method.

**Semi-supervised Recognition.** Pseudo-Label is a basic semi-supervised learning algorithm and can be easily combined with other models. It contains two phases. The first phase is initialization, the recognition model is trained on labeled data. Predictions of the initialized model are assigned on unlabeled data, i.e. pseudo labels. The initialized model is then trained or fine-tuned on the combination of labeled and unlabeled data. In practice, we combine Pseudo-Label method with the two long-tailed recognition models to create two semi-supervised long-tailed recognition baselines. Pseudo-Label combined with LDAM-DRW is the method used in (Yang & Xu (2020)).

Mean Teacher (Tarvainen & Valpola (2017)) is a well-known semi-supervised learning method. It contains a Student model that is trained with SGD and a Teacher model that is updated with moving average of the Student. It is, however, unclear how to train it with Decoupling. We only implement LDAM loss with Student training.

## 4.4 RESULTS

**CIFAR-10-SSLT** results are shown in Table 1 with imbalance ratio 100 and 1000. Our methods outperforms all other methods in overall accuracy.

Our initialized model is equivalent to Decoupling, which shows the worst performance among all methods. Alternate learning improves the overall performance more than $7\%$ when imbalance factor is 100, and $20\%$ with imbalance factor 1000. Most of the improvement is from medium and few-shot classes. The larger improvement on the more imbalanced distribution shows that our method is more effective with more skewed dataset.

When Pseudo-Label is added upon Decoupling, around $5\%$ improvement is achieved with imbalance factor 100. But this improvement diminishes when the data is more imbalanced. This implies the fact that Pseudo-Label is more sensitive to bad tail class labelling.

With the improvement upon Pseudo-Label, our method has the same amount of training epochs on unsupervised data. The extra calculation in our methods compared to Pseudo-Label is from Stage 1 and 2. However, the classifier training is only on supervised data, and only the linear classifier is updated. And label assignment does not involve any back-propagation. The extra time on these two stages are trivial compared to the training of the whole model on the whole dataset.

LDAM-DRW provides very competitive results without any semi-supervised learning methods when imbalance factor is 100. However, it scales up bad when combined with semi-supervised techniques. By adding Pseudo-Label, it only improves $2\%$ of overall accuracy. After looking at the splits results, we find that it improves the few-shot performance at the cost of many-shot. We believe this is because the wrong balancing factor introduced in LDAM loss. It does not match the true distribution, and skews the training process. Mean Teacher makes little difference from Pseudo-Label on LDAM-DRW.

**ImageNet-SSLT** results are shown in Table 1. Our methods outperforms all baseline methods with both ResNet-18 and -50 architectures. The ImageNet-SSLT setting is really challenging that all of the methods give below $30\%$ overall accuracy. In fact, our method is the only one that improves the few-shot performance while maintains the many-shot accuracy.

On ImageNet-SSLT, Pseudo-Label based methods lose efficacy, because it improves few-shot performance with sacrifice on many-shot. This sacrifice is sometimes big, such as Pseudo-Label+LDAM-DRW with ResNet-18. This is not observed when Pseudo-Label is used on CIFAR-10-SSLT, where it improves the many-shot performance. This may be due to the bad many-shot pseudo-label quality on ImageNet-SSLT. Unlike CIFAR-10-SSLT, where the initialized model has $90\%$ of accuracy on many-shot, many-shot performance on ImageNet-SSLT is only around $50\%$. These wrong labels can mislead the training and lower the performance of Pseudo-Label methods. Our method, on the other hand, updates the pseudo labels iteratively, and is less prone to this problem.

Specifically, adding Pseudo-Label on LDAM-DRW decreases the overall performance. This can be explained by the fact that the balancing factor in it does not match the true distribution. Mean Teacher improves upon LDAM-DRW when ResNet-50 is used. But it is still not as good as ours.

Table 2: Ablation results(Accuracy in %) on CIFAR-10-SSLT, Imbalance factor 100 is used. Sampling methods are denoted as R for random, and C for class-balanced. The last two method names shows where the embedding is trained.

| Method | Overall | Many-Shot | Medium-Shot | Few-Shot |
|---|---|---|---|---|
| R + R | 50.9 | 93.0 | 57.8 | 14.1 |
| C + R | 61.2 | 91.3 | 62.6 | 37.6 |
| C + C | 63.3 | 91.2 | 64.4 | 41.6 |
| $\mathcal{D} \cup \mathcal{U}'$ | 70.1 | 89.6 | 68.7 | 56.5 |
| $\mathcal{D}$ | 63.3 | 91.6 | 61.9 | 43.2 |

## 4.5 ABLATIONS

We further study the training choices of alternate learning. This consists of two parts, i.e. the sampling choices and semi-supervised learning choices. Results on CIFAR-10-SSLT with imbalance factor 100 are listed in Table 2.

**Sampling choice.** Currently, during alternate learning we use random sampling in Stage 2 and class-balance sampling in Stage 3. This is consistent with long-tailed recognition (Kang et al. (2020)). However, other combinations are possible. Results are listed in the first 3 lines of Table 2, with naming format: {sampling in Stage 2}+{sampling in Stage 3}. In method names, "R" stands for random sampling and "C" stands for class-balanced.

None of the 3 alternatives can beat the initialized model (Decoupling). This is expected. When the classifier is randomly trained ("R+R" and "C+R"), the model performs bad on few-shot classes. This will in turn harm the training of embedding by pseudo labels on unsupervised subset. "C+C" trains the feature embedding with class-balanced sampling. However, it is balancing on pseudo labels, which can be wrong. The results show that this balancing yields inferior feature embedding.

**Semi-supervised learning choice.** We train feature embedding with the whole dataset, i.e. $\mathcal{D} \cup \mathcal{U}'$, and the classifier with labeled subset $\mathcal{D}$. Other combinations can also be investigated. The classifier can also be semi-supervise trained, i.e. on $\mathcal{D} \cup \mathcal{U}'$. At the same time, feature embedding is trained with or without $\mathcal{U}'$. We show the results in the last 2 lines of Table 2. In these two experiments, the classifier is always trained on $\mathcal{D} \cup \mathcal{U}'$. The difference is whether $\mathcal{U}'$ is used for embedding learning.

Compared to the regular setting, where the classifier is trained on $\mathcal{D}$, when we train it on $\mathcal{D} \cup \mathcal{U}'$, the performance is slightly lower. This can be explained by the fact that wrong pseudo labels in $\mathcal{U}'$ can be propagated through loops if the classifier is optimized on them. This is especially true for few-shot classes, where the accuracy is low. Because of class-balanced sampling, the impact of few-shot classes is amplified. When compared to Table 1, the main performance drop is from few-shot classes. This confirms our assumption.

However, when we further remove the unsupervised training of embedding, the performance drops a lot. It is even worse than the initialized model (Decoupling). In this case, the feature embedding should be equivalent to that of the initialization. The only difference is the classifier. This further proves the fact that fine-tuning classifier on pseudo-labels harms the performance.

## 5 CONCLUSION

This work introduces the semi-supervised long-tailed recognition problem. It extends the long-tailed problem with unsupervised data. With the property of labeled and unlabeled data obeying the same distribution, this problem setting follows the realistic data collection and annotation workflow.

A method based on alternate learning is proposed. By separating supervised training from semi-supervised and decoupling the sampling methods, it incorporates the decoupling training scheme in long-tailed recognition with semi-supervised learning.

Experiments show that the proposed method outperforms all current baselines. When results are split based on class cardinality, the method exhibits its robustness to defective pseudo labels. This is especially true for few-shot classes.

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

## A   TRAINING DETAIL

In initialization, the feature embedding is trained with 200 epochs, and classifier is learned in 10 epochs after that. Stage 2 contains 40 epochs of fine-tuning of the embedding on the whole dataset. In 5 loops of stages, it is in total 200 epochs of embedding fine-tuning. There are also 10 epochs of classifier fine-tuning in Stage 3 per loop. In semi-supervised learning loss (3), $\lambda = 1$ is used.

SGD optimizer with learning rate of $0.1$ is used with cosine annealing during training in all stages. The momentum is $0.9$, and weight decay is $0.0005$.

All comparison methods are implemented with the hyper-parameters in their papers. The codes from authors are used when available.

Table 3: Pseudo label accuracy on unlabeled training subset. CIFAR-10-SSLT with imbalance ratio 100 is used. Compared to testing set, the unsupervised subset is not balanced. In result, the overall accuracy is higher than that on testing set, because of the domination of many-shot classes. The results in many/medium/few-shot splits are more useful.

| Loop | Overall | Many-Shot | Medium-Shot | Few-Shot |
|---|---|---|---|---|
| 0 | 87.7 | 92.3 | 63.0 | 41.8 |
| 1 | 87.9 | 92.3 | 64.0 | 48.1 |
| 2 | 87.8 | 92.1 | 64.7 | 52.2 |
| 3 | 87.8 | 91.8 | 65.3 | 55.8 |
| 4 | 87.7 | 91.6 | 65.8 | 57.8 |

Table 4: Results(Accuracy in %) on CIFAR-10-SSLT, imbalance factor 100. ResNet-32 is used for all methods.

| Method | Overall | Many-Shot | Medium-Shot | Few-Shot |
|---|---|---|---|---|
| LDAM-DRW | 77.34 | 91.8 | 74.3 | 73.7 |
| Pseudo-Label + L | 81.1 | 87.6 | 75.6 | 80.7 |
| Decoupling | 68.2 | 91.6 | 66.9 | 49.2 |
| Ours | 83.1 | 92.0 | 77.7 | 80.4 |

## B    MORE ABLATIONS

**Accuracy on unsupervised training subset.** In Stage 1, we assign pseudo labels for all samples in $\mathcal{U}$. Table 3 reveals how the accuracy changes along loops in all splits. Few-shot split performance improves much faster than others. This proves the effectiveness of our alternate learning scheme, and explains why our method outperforms the baselines by a large margin in few-shot classes.

The unsupervised subset has a long-tailed distribution, so the overall performance is dominated by many-shot. However, alternate learning still gets benefits from the improvement on few-shot split. Accuracy on different splits is more useful when we analyze how the model evolves during training.

**More results.** Table 5 compares our method to Salsa and FixMatch. The gains are significant for all data splits. And ResNet-32 results are shown in Table 4.

## C    iNATURALIST2018-SSLT

**Dataset.** We further curate a benchmark for semi-supervised long-tailed recognition based on iNaturalist 2018 (Van Horn et al. (2018)). iNaturalist 2018 is a long-tailed dataset sampled from natural distribution. We follow the distribution in both of the labeled and unlabeled subset. More specifically, Samples in each class is randomly down-sampled one-fifth of the total number as labeled data, and the remains are assigned as unsupervised subset. Classes with less than 2 labeled samples are eliminated. In result, iNaturalist2018-SSLT contains 8080 classes, with labeled samples from 200 to 2, and the unsupervised subset is 4 times larger.

Classes are divided into three splits based on the number of labeled samples: many-shot ($[100, +\infty)$), medium-shot ($[10, 100)$), and few-shot ($[2, 10)$). It is a extremely long-tailed dataset, with 134 many-shot classes, 1220 medium-shot classes, and 7010 few-shot classes.

**Results.** Results are shown in Table 6. Our method is the only one that improves the overall performance upon baseline. iNaturalist2018-SSLT is different from our other benchmarks in the amount of few-shot classes. It has a very long tail taking up 87% of the label space. This makes the dataset especially hard when combined with unsupervised data.

With the inferior quality of predictions, we see significant drop of Pseudo-Label method in many-shot split. In fact, Pseudo-Label decreases the accuracy of baselines in all splits. Our method mitigates this problem, and improve the few-shot performance. Given the fact that most classes are in few-shot split, our method is the only one that increase the overall performance.

Table 5: Results(Accuracy in %) on CIFAR-10-SSLT, imbalance factor 100. ResNet-18 is used for all methods.

| Method | Overall | Many-Shot | Medium-Shot | Few-Shot |
|---|---|---|---|---|
| Salsa | 59.6 | 82.5 | 60.7 | 41.5 |
| FixMatch | 64.1 | 83.6 | 62.4 | 50.6 |
| Ours | 71.3 | 89.5 | 67.7 | 60.2 |

Table 6: Results(Accuracy in %) on iNaturalist2018-SSLT. ResNet-50 are used for all methods. For many-shot $t > 100$, for medium-shot $t \in (10, 100]$, and for few-shot $t \leq 10$, where $t$ is the number of labeled samples.

| Method | Overall | Many-Shot | Medium-Shot | Few-Shot |
|---|---|---|---|---|
| Decoupling | 27.9 | 54.1 | 41.7 | 24.8 |
| Pseudo-Label + Decoupling | 26.3 | 39.9 | 35.8 | 24.3 |
| Ours | 28.4 | 49.5 | 38.7 | 26.1 |

**Comparison among benchmarks.** From CIFAR-10-SSLT to ImageNet-SSLT and iNaturalist2018-SSLT, the datasets have more and more classes and few-shot classes. In result, they are more and more challenging. This challenge makes Pseudo-Label method ineffective. From CIFAR-10-SSLT to ImageNet-SSLT, the shortcoming first appears in many-shot splits. On ImageNet-SSLT, Pseudo-Label improves the few-shot performance with a sacrifice of many-shot performance. Our method is more robust to this difficulty. It keeps the many-shot performance while improves the few-shot performance. On iNaturalist2018-SSLT, the Pseudo-Label improvement on few-shot split also disappears, and the drop on many-shot is big. Our method, however, can still improves the few-shot performance and control the drop of many-shot compared to the baseline.

All of these results show that semi-supervised long-tailed recognition is a challenging problem. Given the fact that this problem follows the natural workflow of data collecting, we believe it deserves more attention in the literature.

