# OpenReview forum: "Semi-supervised Long-tailed Recognition using Alternate Sampling"
_ICLR.cc/2022/Conference — ICLR 2022 Submitted_

### Official Review · Reviewer_minL · 2021-11-02

**Correctness:** 3
**Technical Novelty And Significance:** 3
**Empirical Novelty And Significance:** 3
**Recommendation:** 5
**Confidence:** 3

**Main Review:**

Strength:

1. The idea of alternating the sampling strategy is very simple yet powerful.
2. The paper identifies an important problem for using pseudo-label in long-tailed learning: the problem with inaccurate pseudo-labels exacerbates when employing class-balanced sampling. The finding comes with solid empirical evidence (Table 2) and analysis (Sect 3.5).
3. The empirical results are promising. The proposed method outperforms the existing methods by a significant margin (Table 1). Sufficient ablation studies are provided. Particularly, I find the loop-wise pseudo label accuracy shown in Table 3 insightful.


Weakness:

1. A semi-supervised feature learning baseline is missing.

   This is my main concern about the paper. The key argument in the paper is that feature learning and classifier learning should 1) be decoupled, 2) use random sampling and class-balanced sampling respectively, 3) train on all labels and only ground-truth labels respectively. The authors, therefore, propose a carefully designed alternate sampling strategy.

   However, a more straightforward strategy could be 1) train a feature extractor ($f$ in the paper) and a classifier ($g\prime$ in the paper) using random sampling and any semi-supervised learning method on all data, then 2) freeze the feature extractor ($f$) and train a new classifier ($g$ in the paper) using class-balanced sampling on data with ground-truth labels. Compared with the alternate sampling strategy proposed in this paper, the semi-supervised feature learning baseline will take less implementation effort and is easier to combine with any semi-supervised learning methods.

   The baseline seems to be missing in the paper. Although the naive baseline may not give the best performance, it should be compared to justify the sophisticatedly designed alternate sampling strategy.

2. References are not up-to-date.

   All references in the paper are in or before 2020. In fact, much research progress has been made since then. For example, some recent works [1, 2] study the class-imbalanced semi-supervised learning as well, a discussion on these methods should be necessary. A recent survey on long-tailed learning [3] could be a useful resource to help update the related works in the paper.

3. Minor issues: "The model is the fine-tuned on the combination of ..." -> "The model is then fine-tuned on the combination of ..."

[1] Su et al., A Realistic Evaluation of Semi-Supervised Learning for Fine-Grained Classification, CVPR 2021

[2] Wei et al., CReST: A Class-Rebalancing Self-Training Framework for Imbalanced Semi-Supervised Learning, CVPR 2021

[3] Deep Long-Tailed Learning: A Survey, arXiv 2021

**Summary Of The Paper:**

This paper proposes to alternate the sampling strategy in the semi-supervised setting and therefore fully exploit the advantage of decoupled learning.

**Summary Of The Review:**

My overall attitude towards the paper is borderline. I appreciate the simplicity and effectiveness of the method provided in the paper. However, I believe that a more straightforward baseline should be compared in order to justify the design of the alternate sampling strategy.

---

### Official Review · Reviewer_PFkd · 2021-11-03

**Correctness:** 2
**Technical Novelty And Significance:** 2
**Empirical Novelty And Significance:** 2
**Recommendation:** 5
**Confidence:** 5

**Main Review:**

The biggest weakness is the experiments. I am not sure why the authors choose ResNet18 as the backbone for Cifar10 experiments. In previous papers, people are using ResNet32, and even without the help of unlabeled data, the results on Cifar10-LT (imbalance 100) were already over 70% using ResNet32 (for example the baselines, not the actual method, reported in this paper [1]). It is hard to judge how good the proposed method is by the reported results, especially when the compared baselines are limited. Most of the existing long-tail methods after decoupling are not compared in the experiments (for example, BBN[2], logit-adjustment[3], and RIDE[4]). Although these methods were not designed for semi-supervised learning, simply adding a pseudo-label component to these baselines is not complicated. In addition, advanced semi-supervised learning methods (like FixMatch[5]) are not discussed either.

Besides the experiments, I think the proposed method lacks novelty as well. Simply adding a pseudo-label component to the decoupling training procedure is too engineering. Since this paper is not a theoretical paper, the novelty of the method is a little more important.


[1] Cai, J., Wang, Y., & Hwang, J. N. (2021). ACE: Ally Complementary Experts for Solving Long-Tailed Recognition in One-Shot. In Proceedings of the IEEE/CVF International Conference on Computer Vision (pp. 112-121).

[2] Zhou, B., Cui, Q., Wei, X. S., & Chen, Z. M. (2020). Bbn: Bilateral-branch network with cumulative learning for long-tailed visual recognition. In Proceedings of the IEEE/CVF Conference on Computer Vision and Pattern Recognition (pp. 9719-9728).

[3] Menon, A. K., Jayasumana, S., Rawat, A. S., Jain, H., Veit, A., & Kumar, S. (2020). Long-tail learning via logit adjustment. arXiv preprint arXiv:2007.07314.

[4] Wang, X., Lian, L., Miao, Z., Liu, Z., & Yu, S. X. (2020). Long-tailed recognition by routing diverse distribution-aware experts. arXiv preprint arXiv:2010.01809.

[5] Sohn, K., Berthelot, D., Li, C. L., Zhang, Z., Carlini, N., Cubuk, E. D., ... & Raffel, C. (2020). Fixmatch: Simplifying semi-supervised learning with consistency and confidence. arXiv preprint arXiv:2001.07685.

**Summary Of The Paper:**

This paper proposes a long-tailed semi-supervised learning setting where both labeled and unlabeled data from a long-tailed distribution exist. To address this setting, they use an iterative decoupling training method, which is based on Kang et al 2020. The only difference is after initialization using the original decoupling method, pseudo labels are obtained from unlabeled data and the feature extractor is updated with pseudo labels, then the classifier is updated using the new feature representations. Two new benchmarks are provided, and the methods had improved performance over selected baselines.

**Summary Of The Review:**

I think the proposed setting is interesting but the experiments are weak. Since the paper is not a theoretical paper, I think the novelty of methods should also be considered. However, the proposed method lacks sufficient novelty since the only additional component is a pseudo-label generation with a consistency loss.

---

### Official Review · Reviewer_Y7tQ · 2021-11-03

**Correctness:** 3
**Technical Novelty And Significance:** 2
**Empirical Novelty And Significance:** 1
**Recommendation:** 1
**Confidence:** 5

**Main Review:**

Strength
- This paper is easy to read, and the proposed method is easy to follow.

Weakness
1. The proposed framework is not new. Semi-supervised long-tailed recognition has already been studied with the name of imbalanced semi-supervised learning [1,2,3].

2. The proposed method is also not novel. In Eq. (2), the authors propose to use temporal consistency loss for learning unlabeled data, which is similar to the method of [5]. This is already outdated in the semi-supervised learning literature where better loss functions [6,7] or pseudo-labeling-based methods [8] show the state-of-the-art performances.

3. The main technical contribution, alternating sampling is also nothing but simple engineering that any long-tailed recognition or semi-supervised learning researchers do. In order to claim the significance of the proposed method, the authors should have at least provided a more comprehensive analysis on the proposed method (e.g., with different imbalance factors).

4. The experiment is also too weak. The authors should have evaluated the proposed method on more training setups (at least one more imbalance factor) and more realistic datasets (iNaturalist [9] or other large-scale datasets) compared with stronger baseline methods (imbalanced semi-supervised learning [2,3], recent long-tailed recognition [10], or at least the FixMatch [8]).

[1] Hyun et al., Class-imbalanced semi-supervised learning. arXiv 2020.

[2] Kim et al., Distribution aligning refinery of pseudo-label for imbalanced semi-supervised learning. NeurIPS 2020.

[3] Wei et al. CReST: A Class-Rebalancing Self-Training Framework for Imbalanced Semi-Supervised Learning. CVPR 2021

[4] Oh et al., Distribution-Aware Semantics-Oriented Pseudo-label for Imbalanced Semi-Supervised Learning. arXiv 21.

[5] Laine et al., Temporal Ensembling for Semi-Supervised Learning. ICLR 2017

[6] Tarvainen et al., Mean teachers are better role models: Weight-averaged consistency targets improve semi-supervised deep learning results. NIPS 2017

[7] Berthelot et al., Mixmatch: A holistic approach to semi-supervised learning.. NeurIPS 2019.

[8] Sohn et la., Fixmatch: Simplifying semi-supervised learning with consistency and confidence. NeurIPS 2020.

[9] Su et al., The Semi-Supervised iNaturalist-Aves Challenge at FGVC7 Workshop. arXiv 2021

[10] Menon et al., Long-tail learning via logit adjustment. ICLR 2021.


**Summary Of The Paper:**

The authors tackle the problem of semi-supervised long-tailed recognition. To address the semi-supervised long-tailed recognition problem, the authors introduce an alternating sampling framework.

**Summary Of The Review:**

There are various obvious weaknesses in this paper including the comprehensiveness of the experiments and the novelty of the problem definition and the proposed method. Therefore, recommend the authors significantly revise and re-submit this paper to the other venue.

---

### Official Review · Reviewer_mLKc · 2021-11-03

**Correctness:** 3
**Technical Novelty And Significance:** 2
**Empirical Novelty And Significance:** 2
**Recommendation:** 5
**Confidence:** 5

**Main Review:**

Strengths:
1. The author propose the new setting of semi-supervised long-tailed recognition, where both labelled and unlabelled data are long-tailed.
2. This paper is well-written, clear and easy to understand. The structure of this paper is also good.
3. The proposed 3-staged method seems reasonable. I am not sure if the author is based on the findings that some balanced strategy, like class-balanced sampling, will hurt the quality of learned feature. Even they achieve better results. The improvement lies in classifier. Thus the  authors update the feature and classifier separately.

Weaknesses:
1. It is also about the new setting proposed. It is actually more like considering the characteristics of long tail in semi-supervised learning. It may not so significant by constraining that the unlabelled data also exhibit long-tailed distribution in long-tailed recognition. As claimed, the unlabelled data will help our classification, that means we may use as many as unlabelled data as possible, for example, the open-set data.

2. Even we consider the long tailed case in unlabelled data, should we consider different long tailed versions? for example, the unlabelled data maybe uniformly distributed? The degree of skewness is less or more severe than the labelled data? Even worse, the many-, medium-, few- classes in unlabelled data may shift since the collected labelled data are easily biased.

3. In Table 1, the overall results are the best. However, the few- accuracy of proposed methods seems lower than "Pseudo-Label + L" and "Mean-teacher+L". Why is the case? One motivation of proposed method is trying to improve the result when predicting the label of unlabelled data. It seems contradictory with the motivation. The proposed method did not improve the result of few-classes.



**Summary Of The Paper:**

This paper propose a new setting, named semi-supervised long-tailed recognition. They consider both of the labelled data and unlabelled data exhibit long-tailed distribution. To solve the problem, they propose an alternate sampling framework which learns the feature and classifier separately and update them iteratively. On two datasets, the results validate the efficacy of proposed method.

**Summary Of The Review:**

The author proposes a new setting of semi-supervised long-tailed classification and clearly demonstrated it. However, the new setting seems minor and not so novel. I thus give the score of 5.

---

### Official Review · Reviewer_6A7n · 2021-11-04

**Correctness:** 4
**Technical Novelty And Significance:** 3
**Empirical Novelty And Significance:** 2
**Recommendation:** 5
**Confidence:** 4

**Main Review:**

Pros:
1)	I believe this work is well motivated. The setting is more realistic than supervised imbalanced recognition in that imbalance usually comes together with unlabeled datasets. In this case, randomly sampling a part of the examples and labeling them manually is reasonable. In this sense, the proposed setting is more closed to what practitioners will encounter in the wild.
2)	The manuscript is overall well-written and easy to follow. The authors introduce the setting and the motivations clearly. The contributions are also clearly summarized.
3)	The intuition behind each stage of the method is stated clearly and emphasized. From my perspective, the proposed method is not a simple combination of pseudo-labeling with decoupling since the authors managed to associate each part of the networks with different training stages and take the effect of imbalance on pseudo-labels into consideration.

Cons:
1)	The baselines are not strong enough. Although the proposed method outperforms the baselines by a large margin, it is still not clear how good the performance is. For supervised imbalanced recognition, the authors consider LDAM+DRW and decoupling as baselines. For standard semi-supervised learning, the authors compare with pseudo-labeling and mean teacher. However, these are not recent state-of-the-arts. It would be better to compare with leading methods on https://paperswithcode.com/sota/long-tail-learning-on-imagenet-lt and better semi-supervised methods such Fixmatch and Noisy student training.
2)	It is unclear how the pseudo-labels are refined by stages 2 and 3. The authors should provide numbers or figures to demonstrate the effect of stages 2 and 3 separately.
3)	For this setting, a more natural method could be: train the representations on all data with self-supervised learning, and fine-tune the representations with the supervised data, and train the final classifier with class aware-sampling. It could e interesting to compare with this baseline.


**Summary Of The Paper:**

This paper proposes a new setting--semi-supervised long-tailed recognition. To harness the imbalanced unlabeled data, the authors combined the decoupling in long-tailed recognition and pseudo-labeling in semi-supervised learning, which formulates a three-stage method. Stage 1 generates pseudo-labels with a classifier trained with class-aware resampling. Stage 2 fine-tunes the feature extractor on with pseudo-labeling. Stage 3 trains the classifier with class-aware resampling on top of the refined feature extractor. Experiments indicate that the proposed method outperforms baselines.

**Summary Of The Review:**

I like the setting and the method, but I still lean towards rejecting the paper, because the experiments are not convincing enough.

---

### Decision · Program_Chairs · 2022-01-20

**Decision:**

Reject

**Comment:**

The paper addresses semi-supervised learning with unbalanced class distribution, a.k.a long-tail. The main idea is to alternate learning of the representation and the classifier.
Reviewers pointed out that several papers already addressed this learning setup, often under the name "imbalanced semi-supervised learning". No rebuttal was submitted.

The paper should make direct comparison to recent papers listed by reviewers, both in terms of the technical approach and in terms of empirical experiments. It cannot be accepted tot ICLR.